# Snakebites in "Invisible Populations": A cross-sectional survey in riverine populations in the remote western Brazilian Amazon

Guilherme Kemeron Maciel Salazar[1,2], Joseir Saturnino Cristino[1,2], Alexandre Vilhena Silva-Neto[1,2], Altair Seabra Farias[1,2], João Arthur Alcântara[1,2], Vinícius Azevedo Machado[1], Felipe Murta[2], Vanderson Souza Sampaio[1,2,3], Fernando Val[1,2], André Sachett[1,2], Paulo Sérgio Bernarde[4], Marcus Lacerda[1,2,5], Fan Hui Wen[6], Wuelton Monteiro[1,2⍟]*, Jacqueline Sachett[1,7⍟]*

1 Escola Superior de Ciências da Saúde, Universidade do Estado do Amazonas, Manaus, Brazil, 2 Diretoria de Ensino e Pesquisa, Fundação de Medicina Tropical Dr. Heitor Vieira Dourado, Manaus, Brazil, 3 Sala de Análise de Situação em Saúde, Fundação de Vigilância em Saúde do Amazonas, Manaus, Brazil, 4 Campus Floresta, Universidade Federal do Acre, Cruzeiro do Sul, Brazil, 5 Instituto Leônidas & Maria Deane, Fundação Oswaldo Cruz, Manaus, Brazil, 6 Instituto Butantan, São Paulo, São Paulo, Brazil, 7 Diretoria de Ensino e Pesquisa, Fundação Alfredo da Matta, Manaus, Brazil

⍟ These authors contributed equally to this work.
* wueltonmm@gmail.com (WM); jac.sachett@gmail.com (JS)

**Data Availability Statement:** All relevant data are within the manuscript and its Supporting Information files.

## Abstract

In the Brazilian Amazon, long distances, low healthcare coverage, common use of ineffective or deleterious self-care practices, and resistance to seeking medical assistance have an impact on access to antivenom treatment. This study aimed to estimate snakebite underreporting, and analyze barriers that prevent victims from obtaining healthcare in communities located in 15 municipalities on the banks of the Solimões, Juruá and Purus Rivers, in the remote Western Brazilian Amazon. Information on the participants' demographics, previous snakebites, access to healthcare, time taken to reach medical assistance, use of self-care practices, and the reason for not accessing healthcare were collected through semi-structured interviews. In the case of deaths, information was collected by interviewing parents, relatives or acquaintances. A total of 172 participants who reported having suffered snakebites during their lifetime were interviewed. A total of 73 different treatment procedures was reported by 65.1% of the participants. Participants living in different river basins share few self-care procedures that use traditional medicine, and 91 (52.9%) participants reported that they had access to healthcare. Living in communities along the Juruá River [OR = 12.6 (95% CI = 3.2–49.7; p<0.001)] and the use of traditional medicine [OR = 11.6 (95% CI = 3.4–39.8; p<0.001)] were variables that were independently associated to the lack of access to healthcare. The main reasons for not accessing healthcare were the pprioritization of traditional treatments (70.4%), and the failure to recognize the situation as being potentially severe (50.6%). Four deaths from complications arising from the snakebite were reported, and three of these were from communities on the banks of the Juruá River. Only one of these received medical assistance. We found an unexpectedly high underreporting of snakebite cases and associated deaths. Snakebite victims utilized three main different healing systems: 1) self-care using miscellaneous techniques; 2) official

**Funding:** This research was funded by Fundação de Amparo à Pesquisa do Estado do Amazonas - FAPEAM (PAREV 007/2019, to JS, and PRÓ-ESTADO and POSGRAD calls, to WM) and by the Ministry of Health, Brazil (proposal no. 733781/19-035, to ML). ML (308748/2017-4), PSB (311509/2020-7)), and WM (309207/2020-7) are research fellows from CNPq. The funders had no role in study design, data collection and analysis, decision to publish, or preparation of the manuscript.

**Competing interests:** The authors have declared that no competing interests exist.

medical healthcare generally combined with traditional practices; and 3) self-care using traditional practices combined with Western medicines. To mitigate snakebite burden in the Brazilian Amazon, an innovative intervention that would optimize timely delivery of care, including antivenom distribution among existing community healthcare centers, is needed.

## Author summary

Many patients bitten by snakes in the Brazilian Amazon do not seek medical care since they live great distinces from health facilities and often do not have the financial resources to travel in search of assistance. In this situation of vulnerability, a wide variety of traditional methods, without proven efficacy, and some known to be harmful, are used by individuals. As a result, many cases of snakebites that are not treated in the official health network are consequently not reported to the epidemiological surveillance system, thus generating underreporting. Knowing the proportion of underreported cases and their hostspots is very important for planning interventions that will improve the coverage of the healthcare network and the logistics of delivery of antivenoms. In a pioneering way in Brazil, this study was carried out to estimate snakebite underreporting and analyze obstacles that prevent victims from obtaining healthcare in the communities located in 15 municipalities on the banks of the Solimões, Juruá and Purus Rivers, in the remote Western Brazilian Amazon. Cases of deaths due to snakebites were also investigated. A total of 172 participants who reported having suffered snakebites during their lifetime were interviewed, as well as the circumstances of 4 deaths. Most patients recalled using some traditional medicine in the form of self-care to treat snakebite. In total, there were 73 different treatment procedures, which were quite different between the different regions studied. Almost half of the participants did not seek medical advice, claiming as reason for that the prioritization for traditional treatments and the non-regognition of the situation as potentially severe. In the Juruá River communities, this frequency was around 70%. The preference of traditional medicine was also associated with the lack of access to healthcare. We believe that this situation will only be mitigated if the antivenoms are made available closer to snakebite victims, i.e., in riverside and rural health units.

## Introduction

Snakebite envenomings occur through the inoculation of toxins by several species of snakes, and lead to local and systemic manifestations, as well as potential physical and psychological disabilities, which can have enormous social and economic repercussions. Since the consequences of snakebites are greater in tropical and subtropical regions with less economic development, these were included by the World Health Organization in its list of Neglected Tropical Diseases [1,2]. Envenomings caused by snakes occur mostly in rural areas, which have low human socioeconomic indexes, poor housing and sanitation conditions, and lack of comprehensive healthcare [3,4]. Additionally, rural populations with these risk characteristics face serious difficulties in order to receive adequate treatment since access to healthcare units often involves long journeys, combined with a low availability of means of transportation and significant geographical barriers. Self-care procedures based on traditional medicine and resistance to seeking official healthcare services are also key aspects to understanding the invisibility of such populations to public authorities [5]. Deficiencies in the availability of antivenoms

and lack of adequate training of health professionals are factors that compromise patient care, thus favoring the occurrence of severe cases and deaths due to snakebite envenomations [1,2].

The populations living in the Brazilian Amazon, popularly named "caboclos" or "ribeirinhos", are genetically admixed populations that traditionally live in rural riverine areas [6], in a large part of the Amazonian territory [7]. Many caboclo populations live in hard to reach rural areas with little or no medical assistance, education or sanitation services. Caboclos are "invisible" populations because they represent a failure of past government integration efforts, and are still excluded from the developmental agendas, both of the extractive and agribusiness sectors [8]. As a result, available data show low socioeconomic indexes and high rates of illness, especially for infectious and parasitic diseases, and malnutrition, among individuals living in the remote parts of the Amazon [8]. The populations that live in rural or forested areas are more likely to be affected by snakebites, due to activities such as livestock farming, fishing, and other types of hunting and gathering, which occur in the same natural habitats as venomous species (namely common lancehead *Bothrops atrox*, two-striped forest pitviper *B. bilineatus*, bushmasters *Lachesis muta* and coral snakes *Micrurus* spp.) [9,10].

Snakebite envenomation requires antivenom treatment in the first hours after the bite. In the Amazon, however, antivenom treatment is limited to urban areas, and this results in late medical assistance, antivenom misuse and and frequently lack of availability of antivenoms [10]. In this region, the delay in treatment is significantly associated with complications and lethality [11,12], and usually attributed either to the great distances to be traveled by patients in order to reach a healthcare unit where antivenom is available [12], or to cultural factors that cause patients to choose traditional treatment [5,13,14]. Antivenom underuse was reported in rates ranging from 52% to 81% for the severe envenomations caused by pit vipers and bushmasters, respectively [11]. In this region, lethality was significantly associated with a lack of antivenom administration and antivenom underuse [11]. Antivenom underdosing was significantly higher in indigenous populations compared to urban and countryside populations [15,16].

Long distances combined with geographical barriers, such as rivers, and badly maintained roads, have an impact on the itineraries of riverine populations when seeking antivenom treatment. Previous studies report a high fragmentation of the itineraries, which can be marked by several changes of mode of transportation along the route, until arrival at the hospital [5,16]. Likewise, another factor that can affect access to antivenom is related to the cost in terms of travel, pre- and post-hospitalization medications, as well as the loss in income due to time of work lost [5,17]. From a cultural perspective, a large proportion of patients choose to undergo traditional therapeutic resources. Plant and animal-derived preparations, blessings and prayers, as well as self-medication with conventional drugs are commonly used by patients before making the decision to search for the health service [5,13,14]. Traditional practices, such as the use of tourniquets, incisions in the affected area and use of substances of several origins, are believed by the patients to be a way to remove venom or prevent its spread around the body. Use of deleterious self-care practices are recorded across the world as the cause of late medical assistance and poor prognosis in snakebites [5,18,19]. Some of these harmful procedures can lead to complications from snakebite, such as secondary bacterial infections in the affected limb, necrosis and compartment syndrome, and may lead to life-long disabilities that greatly impact on the victim's quality of life [13,19,20].

In this study the barriers that riverine populations face in order to reach healthcare in the remote Western Brazilian Amazon are analyzed and snakebite rates during lifetime are estimated, thus demonstrating the complexity of the accessibility to antivenom treatment for the caboclos.

## Methods

### Ethics statement

The data collection for this study was carried out after approval by the Human Research Ethics Committee of the Amazonas State University (approval number 3.223.046/2019), in compliance with Resolution number 466/2012, of the Brazilian National Health Council. Before starting interviews in a community, the community leader was consulted and gave his approval. All participants signed the consent form before participating in the interviews, and after receiving information about the objectives and procedures of the study. Children and adolescents signed an assent term and their parents or legal guardians signed a consent term agreeing to the minor's inclusion in the study. In the cases of death, after signing a consent form, a family member was interviewed to collect the information.

### Study sites, access to communities and subjects

The state of Amazonas is located in the western Brazilian Amazon, has 62 municipalities and comprises an area of 1,570,946.8 km$^2$ (Fig 1). The estimated population of the state was 4,207,714 inhabitants in 2020, with ~20% living in rural areas. Approximately 52.8% of the population lives

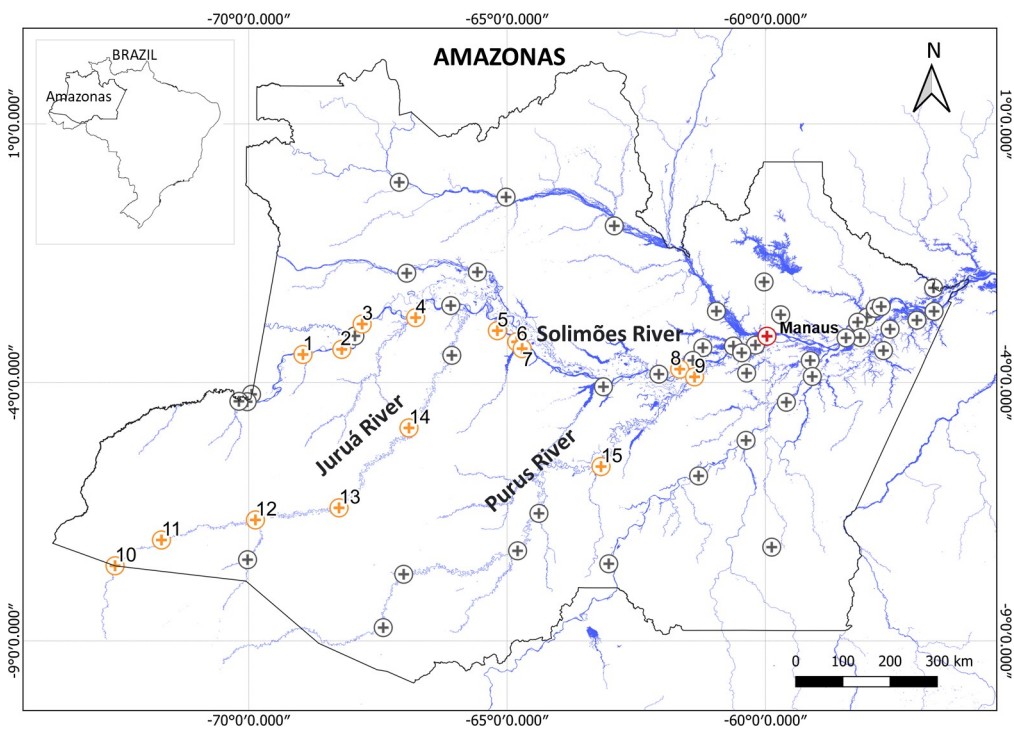

**Fig 1. Location of the state of Amazonas, Wester Brazilian Amazon.** Circles represent urban areas of 61 municipalities that have snake antivenoms available. Yellow circles represent municipalities with at least one riverine community included in this study. A total of 141 communities in 15 municipalities were visited: along the Solimões River (Alvarães, Amaturá, Anori, Jutaí, São Paulo de Olivença, Tefé, Tonantins, and Uarini), the Juruá River (Carauari, Eirunepé, Guajará, Ipixuna, and Itamarati), and the Purus River (Anori, Beruri, and Tapauá). 1-São Paulo de Olivença; 2-Amaturá; 3-Tonantins; 4-Jutaí; 5-Uarini; 6-Alvarães; 7-Tefé; 8-Anori; 9-Beruri; 10-Guajará; 11-Ipixuna; 12-Eirunepé; 13-Itamarati; 14-Carauari; 15-Tapauá. Manaus (red circle) is the state capital and has the referral center for treating severe snakebite cases at the Fundação de Medicina Tropical Dr. Heitor Vieira Dourado. Base used to create map is from the IBGE (Brazilian Institute of Geography and Statistics), which is freely accessible for creative use in shapefile format, in accordance with the Access to Information Law (12,527/2011) (https://portaldepapas.ibge.gov.br/portal.php#homepage).

in the state capital, Manaus [21]. The state has a reduced coverage of highways and roads, with most of the displacement occurring via river transportation. The region is densely covered by an evergreen rainforest, with the upland forests (*terra firme* forest), as well as floodplains (*várzeas*) and flooded areas (*igapós*). The fluvial system present in the state of Amazonas is part of the great Amazon basin, which is the largest body of fresh water in the world, with 3,889,489.6 km$^2$ [22]. This basin is made up of the Amazon River, its tributaries and the lowland lakes that interact with the rivers. Seasonal fluctuation in the water level is an important force function that drives the system ecology. During the period of high water level, the entire system is flooded.

In this study, communities located on the banks of the Solimões, Juruá and Purus rivers were studied (Fig 1). The Solimões River becomes Amazon River after joining the Negro River in Manaus. The Solimões-Amazon waterway is the most important in the Amazon region, serving Brazil, Colombia, Peru, Ecuador and Bolivia. The Juruá and Purus Rivers originate in the Peruvian territory and flow through the territory of Peru and the Brazilian states of Acre and Amazonas. Together, they are two of the major tributaries of the right bank of the Solimões River, playing an extremely important role for the region as a fluvial highway for several communities.

A total of 141 communities were visited in 15 municipalities along the Solimões, Juruá, and Purus Rivers. Access to the communities took place through two field trips. On the first trip, the communities located on the banks of the Juruá and Solimões Rivers were visited between January and March 2019, with the team from the National Metrology Institute (INMETRO), in the Basic River Inspection and Research Unit (UBFFP) (Fig 2A). On the second trip, in October and November 2019, the communities located on the banks of the Purus River were visited, with the Brazilian Navy team, on board the Hospital Assistance Ship "Carlos Chagas" (U-19) (Fig 2B).

These communities present a model of land occupation and natural resource use that is predominantly subsistence-oriented and weakly articulated with the market, based on the intensive use of family manpower, low impact technologies derived from inherited knowledge, and are, as a general rule, sustainable [23]. Population living in this area presents a fragile socio-economical organization, and the main subsistence activities are fishing, agriculture, the extraction of forest resources, hunting, livestock raising, and incipient trading [9,24]. During the visits, cassava plantations were observed in most communities, the production of which serves essentially to prepare cassava flour for their own consumption (Fig 2C). Fishing and game hunting are important sources of food for these populations (Fig 2D). The houses are usually built of wood, and they are made on solid, floating or stilt bases (Figs E, F, G and H). There are also açaí plantations, and fruit and vegetable gardens in most communities (Fig F). Pig and poultry rearing is common, but cattle breeding was rarely seen. The exchange of goods between communities is a common practice. For example, the exchange of dried 'pirarucu' meat (*Arapaima giga*) and game meat for sugar, salt, beans, rice and even some medicines was sometimes seen. In addition, boats that function as floating markets anchored close to these communities were frequently observed. The communities along the banks of the Solimões and Purus Rivers have electricity available, though this is not the case in the communities along the Juruá River. Only the most populated communities have primary schools. None of the communities have health services, therefore, when there is a need for medical care, such as when snakebites occur, patients must travel to the nearest city.

## Study design and sampling

This is a cross-sectional study using a chain-referral (snowball) sampling technique, a non-probability sampling technique in which previously included participants in each community recruited the future subjects from among their acquaintances [25]. Upon arrival in each

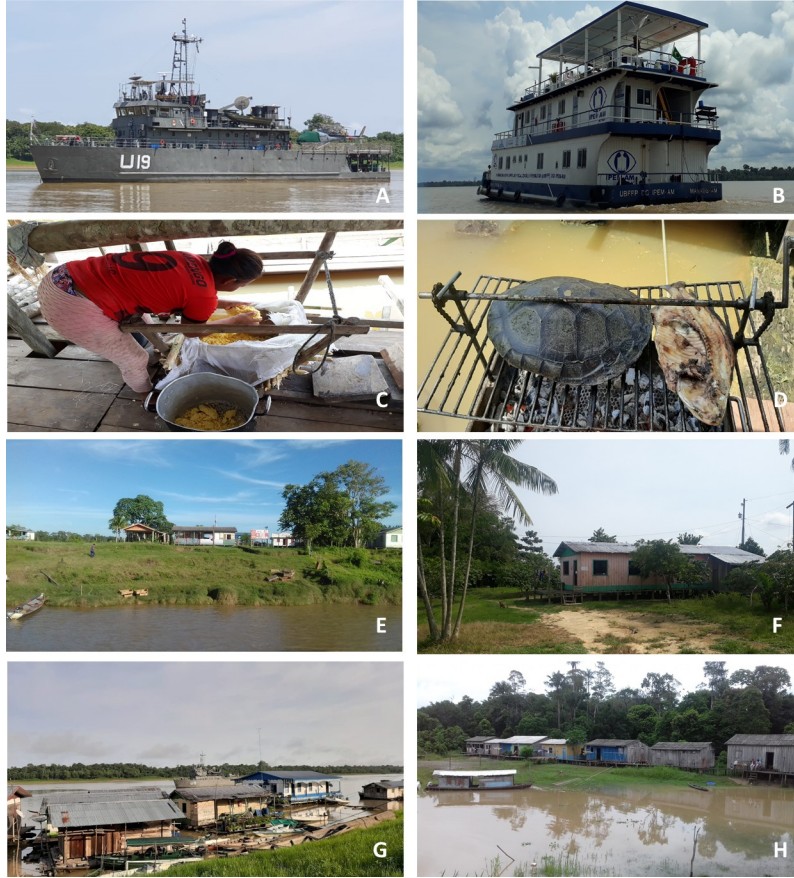

**Fig 2. Means of transport used to access communities and characteristics of the communities in the study area.**
On the first trip, the communities located on the banks of the Juruá and Solimões rivers were visited from January to March 2019. The researchers accompanied the team from the National Metrology Institute (INMETRO), on the Basic River Inspection and Research Unit (UBFFP), which is a vessel that carries out inspection activities and scientific research activities in the Brazilian Amazon (Fig 2A). On the second trip, in October and November 2019, the communities located on the banks of the Purus river were visited along with the Brazilian Navy team on board the Hospital Assistance Ship "Carlos Chagas (U-19)" (Fig 2B). Fig 2C shows artisinal production of manioc flour, one of the subsistence foods and main economic activities of the riverine populations. Fig 2 D shows the preparation of tracajá (*Podocnemis unifilis*), a chelonian considered a delicacy of Amazonian cuisine, hunted and consumed by riverside dwellers, and the tambaqui (*Colossoma macropomum*), a regional fish, both consumed as important sources of protein. Fig E shows a typical wooden house, located on dry land, which is not affected by the flood phase of the rivers. Fig F shows a wooden house built on dry land with several fruit trees in the peridomicile, as well as access trails. Fig G shows floating houses, built on logs of large trees that follow the course of rivers in the flood and ebb phases. The picture also shows several canoes, which are an important means of transport and used for short trips. Fig H shows houses built on stilts (*palafitas*) that are close to the primary forest and interconnected by wooden bridges. Pictures were taken by the first author.

community, the community leader was sought to obtain permission to carry out the survey after explaining the study objectives and methods. The community leader acted as the initial informant to nominate the first potential participant, i.e., a subject with a previous history of snakebite. This participant was then visited at home and, after accepting to participate in the study, responded to the data collection instruments. Finally, this participant indicated, through their social network, another participant who met the eligibility criteria and could potentially contribute to the study. The inclusion of new participants finished once the saturation point had been reached, that is, when the recruited participants no longer had new snakebite cases in the community to indicate to the researcher.

## Data collection

Data were collected through semi-structured interviews by a researcher who is trained in field epidemiology, via the application of a questionnaire with open and closed questions. Children and adolescents were interviewed in the presence of their parents or legal guardians, who assisted them in the responses. In the case of reported deaths, the information regarding the deceased person was collected by interviewing the informant and other adults of the house.

**Socio-demographic variables.** This section of the questionnaire assessed information on demographic and socio-economic characteristics, including gender, age, marital status, education, occupation, monthly income, municipality and community of residence, and characteristics of their houses. The geographical coordinates of each community were collected by the researcher during the visits using GPS (Garmin GPSMAP 64x).

Snakebite history: In this section, participants were asked about the number of snakebites they had suffered, the date of the most recent snakebite, the geographic location where the bite occurred and the work he/she performed, the popular name of the snake that caused the bite, use of protective clothing, the anatomical region of the bite, local and systemic signs and symptoms, use of health services, and the time elapsed from bite to medical care. Additionally, two open questions on self-care procedures (first aid, use of traditional medicines, etc.) were used and, in cases in which the subject was not treated at a health service, the reason given for this. A board containing photographs of snakes was shown to the patient to assess whether they recognized the specimen responsible for the envenomations [11]. When reported by study participants, deaths involving family members or acquaintances living in the community were also recorded.

After data collection, information on first aid in cases of snakebites, with an emphasis on the need to seek immediate medical care, as well as the nearest hospital to be accessed by participants, was provided by the field researcher.

## Data analysis

Comparison of the participants' characteristics, history of snakebites, access to healthcare, and time elapsed to medical assistance, among the communities located on the three river basins was made using Chi-square test (corrected by Fisher's exact test if necessary). Analysis of association was performed to assess the factors associated to lack of '*access to healthcare*', defined here as the completion of the therapeutic itinerary from the moment of the bite to the participant's admission to the referral unit where the proper medical care was given [5]. Although this is a very simplistic definition of access to healthcare by vulnerable groups [26], we believe that this is a reliable way to classify the participants according to the achievement of the expected endpoint for a patient bitten by a venomous snake, i.e., the admission to a heath unit for antivenom treatment. Analysis included the estimates of access to healthcare in the studied population and identification of factors associated to access. The crude Odds Ratios (ORs) with their respective 95% confidence interval (95%CI) was determined considering lack of access to health care as dependent variable. Logistic regression was used for the multivariate analyses and the adjusted ORs with 95% CI were also calculated. All variables associated with the outcomes at a significance level of $p<0.20$ in the univariate analysis were included in the multivariable analysis. Statistical significance was considered if $p<0.05$ in statistical tests. The analysis was performed using STATA software (StataCorp. 2013: Release 13. College Station, TX, USA). Regarding the reason for not seeking medical care, participants' responses were analysed and grouped in categories, discussed among the researchers for consensus.

## Results

### The STROBE checklist is presented in S1 File

Out of all the communities visited, 81 (57.4%) had at least one resident with history of snakebite, in all the 15 municipalities, with eight along the Solimões River (Alvarães, Amaturá, Anori, Jutaí, São Paulo de Olivença, Tefé, Tonantins, and Uarini), five along the Juruá River (Carauari, Eirunepé, Guajará, Ipixuna, and Itamarati), and three along the Purus River (Anori, Beruri, and Tapauá). In 60 of the communities, no cases of snakebites were reported by the community leader. The information, on these occasions, was confirmed with at least two more residents of the community. The list of communities with at least one participant who claimed to have been bitten by a snake and their geographic locations is presented in S2 File.

### Characteristics of the participants

From the 181 indications obtained from the informants, 172 (95%) individuals were encountered in the communities. There was no refusal to provide written consent. Seventy-eight participants were included in the communities along the Juruá River (45.3%), 75 along the Solimões River (43.6%), and 19 along the Purus River (11.1%). Most of the participants were male (87.6%), aged 49-59-years old (43.5%), illiterate or with $\leq$ 4 years of schooling (62.2%). Most of them were involved in agriculture (58.7%) and fishing activities (18%), and married or in stable relationships (69.2%). Most participants had a monthly income <1 minimum wage (94.9%). Pensions (12.8%) and social security payments (*Bolsa Família*) (9.3%) were other income sources informed by the participants. Wooden houses were the main type of housing (93.6%). The literacy rate was significantly lower in participants living on the banks of the Juruá River. The proportion of participants living on houseboats was significantly higher in communities along the Purus River. The other characteristics were similarly distributed between the three river basins. The characteristics of the study participants are presented in Table 1.

### History of snakebites

A total of 22 (12.8%) participants reported $\geq$3 snakebites during lifetime. In the communities along the Solimões and Juruá Rivers, snakebites were mostly associated with agricultural land, while in the Purus River the cases occurred more often on the trails used to access workplaces and on the river margins, during fishing or leisure activities. *Bothrops* (pit vipers) envenomations predominated in the three river basins, ranging from 55.1% in the Juruá River, to 90.7% in the Solimões River, and 89.4% in the Purus River communities. Some participants living in the Juruá River communities reported envenomations by *Bothrops bilineatus* (popularly known as "papagaia", in Portuguese). The proportion of envenomations caused by the *Lachesis* genus (bushmasters) was significantly higher in the Juruá River region (Table 2).

### Self-care using traditional medicine

The use of traditional treatments was reported by 65.1% of the participants interviewed. Frequency was significantly higher in communities located along the Juruá River (76.9%) compared to those along the Solimões River (53.3%) ($p$ = 0.002). In the communities of the Purus River, this frequency was 63.2%, and did not differ significantly from the other river basins ($p$>0.200).

A total of 73 different treatment procedures or combination of treatments were used by the 172 participants to treat snakebites, with 26 in communities of the Solimões River, 40 in Juruá River, and 11 in the Purus River (S3 File). Only three medications were used in more than one

**Table 1. Characteristics of the 172 study participants.**

| Variables | Community location | | | Total (n = 172) Number (%) |
|---|---|---|---|---|
| | Solimões River (n = 75)¶ | Juruá River (n = 78) | Purus River (n = 19) | |
| | Number (%) | Number (%) | Number (%) | |
| **Gender** | | | | |
| Male | 62 (82.7%) | 71 (91.0%) | 16 (84.2%) | 149 (86.6%) |
| **Age groups (years)** | | | | |
| <18 | 6 (8.0%) | 6 (7.7%) | 1 (5.3%) | 13 (7.5%) |
| 19–45 | 33 (44.0%) | 33 (42.3%) | 9 (47.4%) | 75 (43.5%) |
| 46–60 | 19 (25.3%) | 19 (24.4%) | 5 (26.3%) | 43 (25.0%) |
| ≥60 | 17 (22.7%) | 20 (25.6%) | 4 (21.0%) | 41 (23.8%) |
| **Education (years of study)** | | | | |
| Illiterate | 17 (22.7%) | 41 (52.6%)* | 5 (26.4%) | 63 (36.6%) |
| ≤4 | 21 (28.0%) | 16 (20.5%) | 7 (36.8%) | 44 (25.6%) |
| >4 | 37 (49.3%) | 21 (26.9%)* | 7 (36.8%) | 65 (37.8%) |
| **Main occupation** | | | | |
| Agriculture | 56 (74.7%) | 41 (52.6%) | 4 (21.1%) | 101 (58.7%) |
| Fishing | 5 (6.7%) | 15 (19.2%) | 11 (57.9%) | 31 (18.0%) |
| Retired | 8 (10.6%) | 8 (10.3%) | 2 (10.5%) | 18 (10.5%) |
| Others | 6 (8.0%) | 14 (17.9%) | 2 (10.5%) | 22 (12.8%) |
| **Marital status** | | | | |
| Married/stable relationship | 47 (62.7%) | 59 (75.6%) | 13 (70.0%) | 119 (69.2%) |
| Unmarried | 26 (34.7%) | 16 (20.5%) | 6 (30.0%) | 48 (27.9%) |
| Widow | 1 (1.3%) | 2 (2.6%) | 0 (0.0%) | 3 (1.7%) |
| Divorced | 1 (1.3%) | 1 (1.3%) | 0 (0.0%) | 2 (1.2%) |
| **Monthly income (minimum wages)** | | | | |
| <1 | 71 (94.7%) | 73 (93.8%) | 19 (100.0%) | 167 (94.9%) |
| 1–3 | 3 (4.0%) | 3 (3.7%) | 0 (0.0%) | 6 (3.4%) |
| ≥3 | 1 (1.3%) | 2 (2.5%) | 0 (0.0%) | 3 (1.7%) |
| **Income source#** | | | | |
| Pension | 7 (9.3%) | 11 (14.1%) | 4 (21.1%) | 22 (12.8%) |
| *Bolsa Família* | 4 (5.3%) | 2 (2.6%) | 10 (52.6%) | 12 (9.3%) |
| Subsistence income | 64 (85.4%) | 65 (83.3%) | 5 (26.3%) | 141 (81.9%) |
| **House characteristics** | | | | |
| Wooden house | 71 (94.7%) | 78 (100.0%) | 12 (63.2%)*** | 161 (93.6%) |
| Brick-built house | 1 (1.3%) | 0 (0.0%) | 2 (10.5%) | 3 (1.7%) |
| Houseboat | 3 (4.0%) | 0 (0.0%) | 5 (26.3%)** | 8 (4.7%) |

¶ Reference group for statistical comparisons by Chi-square test (corrected by Fisher's exact test if necessary)

*$p<0.05$

**$p<0.005$

***$p<0.0005$.

#*Bolsa Família* is the largest Brazilian cash transfer program in the country, aiming to bring alleviation of immediate poverty and eradicate hunger.

group of communities, suggesting that communities living on different river basins share few self-care procedures that involve traditional medicine. A preparation of unknown composition called *Específico Pessoa*, which is commonly marketed in the Amazon, was used by 35 participants (31.3%) in communities of the three rivers, on its own or in combination with a series of other preparations. Preparations using leaves or roots of the açaí palm tree (*Euterpe oleracea*)

**Table 2. Characteristics of the 172 study participants according to their history of snakebites.**

| Variables | Community location | | | Total (n = 172) Number (%) |
|---|---|---|---|---|
| | Solimões River (n = 75)¶ | Juruá River (n = 78) | Purus River (n = 19) | |
| | Number (%) | Number (%) | Number (%) | |
| **Number of snakebites suffered** | | | | |
| 1 | 51 (68.0%) | 53 (68.0%) | 15 (78.9%) | 119 (69.2%) |
| 2 | 11 (14.7%) | 16 (20.5%) | 4 (21.1%) | 31 (18.0%) |
| ≥3 | 13 (17.3%) | 9 (11.5%) | 0 (0.0%) | 22 (12.8%) |
| **Date of the last snakebite#** | | | | |
| <3 months | 3 (4.0%) | 2 (2.6%) | 2 (10.5%) | 7 (4.1%) |
| 3–6 months | 1 (1.3%) | 1 (1.3%) | 1 (5.3%) | 3 (1.7%) |
| 6 months-1 year | 5 (6.7%) | 5 (6.4%) | 1 (5.3%) | 11 (6.4%) |
| 1–5 years | 19 (25.3%) | 23 (29.5%) | 3 (15.8%) | 45 (26.2%) |
| 6–10 years | 7 (9.3%) | 10 (12.8%) | 2 (10.5%) | 19 (11.0%) |
| ≥10 years | 24 (32.0%) | 36 (42.6%) | 6 (31.6%) | 66 (38.4%) |
| **Place where snakebite occurred** | | | | |
| Agricultural land | 33 (44.0%) | 17 (21.7%) | 2 (10.5%)*** | 52 (30.2%) |
| Household area | 20 (26.7%) | 26 (33.3%) | 4 (25.1%) | 50 (29.1%) |
| Trails to access workplaces | 1 (1.3%) | 7 (9.0%)** | 7 (36.8%)** | 15 (8.7%) |
| River margins | 7 (9.4%) | 12 (15.4%) | 6 (31.6%)*** | 25 (14.5%) |
| Açaí plantation | 1 (1.3%) | 2 (2.6%) | 0 (0.0%) | 3 (1.7%) |
| Rubber plantation | 0 (0.0%) | 13 (16.7%) | 0 (0.0%) | 13 (7.6%) |
| Hunting | 1 (1.3%) | 1 (1.3%) | 0 (0.0%) | 2 (1.2%) |
| Not remembered | 12 (16.0%) | 0 (0.0%) | 0 (0.0%) | 12 (7.0%) |
| **Use of individual protection** | | | | |
| Yes | 10 (26.8%) | 12 (25.0%) | 3 (20.0%) | 25 (14.5%) |
| **Type of envenomation** | | | | |
| *Bothrops* | 68 (90.7%) | 43 (55.1%)*** | 17 (89.4%) | 128 (74.4%) |
| *Lachesis* | 6 (8.0%) | 34 (43.6%)*** | 1 (5.3%) | 41 (23.8%) |
| *Micrurus* | 1 (1.3%) | 1 (1.3%) | 1 (5.3%) | 3 (1.8%) |
| **Anatomical region of the bite** | | | | |
| Lower limbs | 64 (85.3%) | 70 (89.7%) | 16 (84.2%) | 150 (87.2%) |
| Upper limbs | 11 (14.7%) | 7 (9.0%) | 3 (15.8%) | 21 (12.2%) |
| Others | 0 (0.0%) | 1 (1.3%) | 0 (0.0%) | 1 (0.6%) |

¶ Reference group for statistical comparisons by Chi-square test (corrected by Fisher's exact test if necessary)

# 21 (12.2%) patients did not remember

*$p<0.05$

**$p<0.005$

***$p<0.0005$.

were used by 23 participants (20.5%), 20 in the Juruá River (33.3%) and three in the Solimões River (7.5%). A preparation reported by 15 participants of the Juruá river (25%), was the 'second step tea', prepared from the material scraped from the surfaces of the second step of a wooden staircase, which generally gives access from the river to the community. Black stone was used by nine participants (8%), with 8 in the Juruá River (13.3%) and one in the Solimões River. Most of the participants used plant-based medicines, especially infusions. Some animal-based preparations that used paca, snake, tortoise, caiman, mosquitos, fishes and frog organs were also reported. Nine participants reported the use of conventional medicines, especially

intravenous benzylpenicillin and painkillers (metamizole, acetylsalicylic acid, paracetamol, and diclofenac). Some foods, such as cow's milk and broth made with tortoise meat, were also used for treatment purposes.

### Access to healthcare and associated factors

A total of 91 (52.9%) participants reported that they had access to healthcare and received assistance in a hospital in the urban area of the municipalities. Frequency was similar in the communities along the Solimões and Purus Rivers, with 72 and 74%, respectively. Along the Juruá River, the frequency of access to healthcare was significantly lower (29.5%; $p<0.0005$) (Fig 3A).

Most of the participants who had access to healthcare, sought it in less than 6 hours (65.9%). This frequency was lower in the Purus River communities (35.7%; $p<0.005$). Eleven (12.1%) participants took more than 72 hours to get to the healthcare unit (Fig 3B).

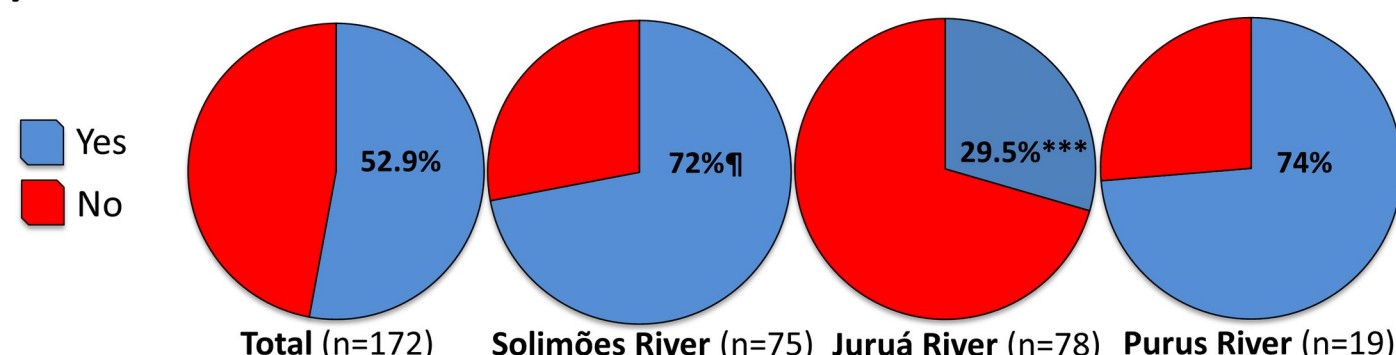

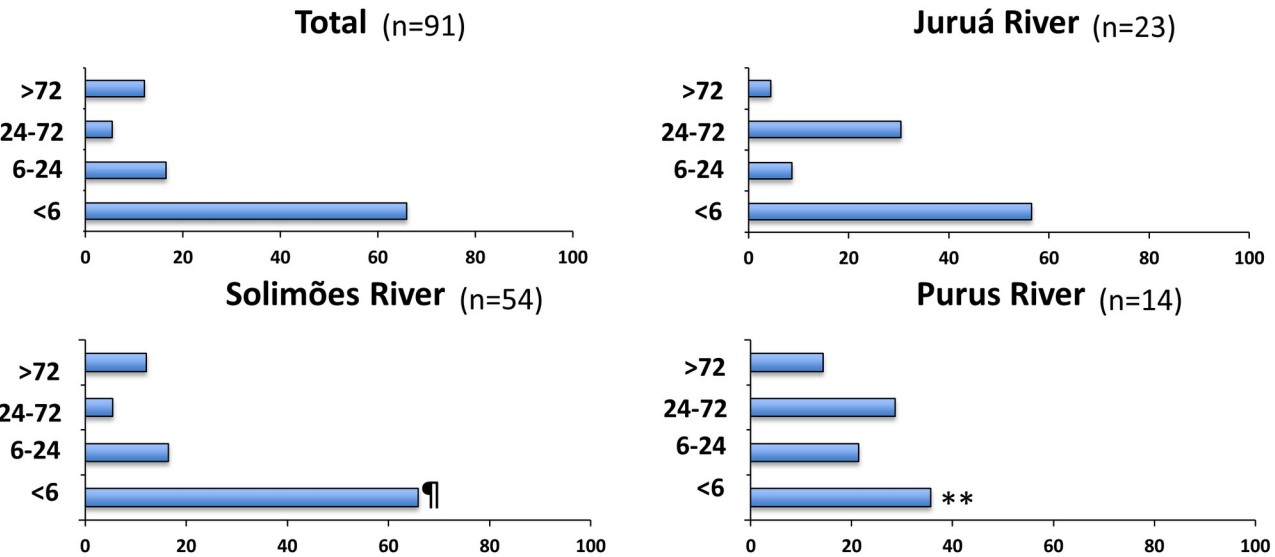

**Fig 3.** Access to healthcare (A) and time elapsed from snakebite to healthcare (hours) (B) according to three river basins. ¶ Reference group for statistical comparisons by Chi-square test (corrected by Fisher's exact test if necessary); *$p<0.05$; **$p<0.005$; ***$p<0.0005$; § Information was not provided by two participants.

In the multivariate analysis, living in communities along the Juruá River [OR = 12.6 (IC 95%CI = 3.2–49.7; p<0.001)] and the use of traditional medicine [OR = 11.6 (IC 95%CI = 3.4–39.8; p<0.001)] were independently associated to the lack of access to healthcare (Table 3).

The main reason for not accessing healthcare as stated by the participants was the prioritization of traditional treatments, relying on their effectiveness (70.4%), followed by failure to recognize the situation as potentially serious (50.6%), lack of financial resources and means of transportation (37%), and resistance to seeking medical assistance (17.3%). Reasons for not accessing healthcare were similar between river basins (Fig 4A). Most participants cited more than one reason for not seeking medical attention, with a more visible overlap between prioritization of traditional treatments and failure to recognize the situation as potentially serious, and prioritization of traditional treatments and lack of financial resources and means of transportation (Fig 4B).

## Deaths from snakebites

In the survey, four deaths from complications arising from the snakebite were informed (Table 4) resulting in a case-fatality rate of 2.3% (4/176). In the Juruá River communities, the estimated case-fatality rate was 3.7% (3/81). The characteristics of the four deaths from snakebite, as described by family members, are presented in Table 3. In summary, three patients lived in communities of the Juruá River and ages ranged from 3 to 81 years old. The adults were bitten during work activities. Only one patient sought medical assistance with the help of a health worker who was visiting the community using a motorboat belonging to the municipality.

## Discussion

### Invisible populations, hidden burden of snakebites

Some surveys performed in riverine communities in the Amazon region have shown a lack of resolution of the transition process of replacement of the common infectious diseases by non-communicable chronic diseases. In this state of mixed morbidity, these riverine populations are exposed to high burdens of undernutrition [27,28] and infectious diseases [29–35], together with obesity, diabetes and hypertension [36,37]. This scenario is caused by social and ethnical inequities, large geographic distances and an inadequate healthcare network, which noticeably limits access to health services in non-urban areas [38,39]. In the anthropological literature, Amazonian caboclos are characterized as invisible populations based on four main reasons: the idealization of the Amazonian landscape as strictly natural; caboclos were never incorporated into the formal economy; Amazonia as a frontier territory; and the fact that caboclo agrarian systems are neo-colonial 'experiments', significantly based on immigrant practices [7]. Since caboclos belong to an essentially informal economy, they are excluded from the developmentalist project of highly capitalized extractivist industries [40]. In this study, we found a population with very low education and income that is deeply dependent on family agriculture and extractive activities for their subsistence, and that has a precarious participation of the state in providing of health services.

Nearly half of the participants reported that they did not have access to a hospital in the urban area of the municipalities where they could receive proper healthcare. This result is similar to that found for Amerindian populations in the Brazilian Amazon, for which only 54% received antivenom treatment [16]. The proportion of participants who did not have access to healthcare was alarmingly high in the communities of Juruá River, and reached 70.5% of the cases. Lack of antivenom administration was independently associated to case fatality in the Brazilian Amazon [11]. Communities located on the Juruá River had significantly less access to healthcare, corroborating the situation of greater exclusion of this population from

**Table 3. Factors associated to access to lack of healthcare in participants living in three river basins, western Brazilian Amazon.**

| Variable | OR | 95%CI | p | aOR | 95%CI | p |
|---|---|---|---|---|---|---|
| **Community location** | | | | | | |
| Solimões River | 1 | . | . | 1 | . | . |
| Juruá River | 6.1 | 3.1–12.4 | 0 | **12.6** | **3.2–49.7** | **<0.001** |
| Purus River | 0.9 | 0.3–2.9 | 0.883 | 0.1 | 0.0–1.6 | 0.115 |
| **Distance from community to the urban area (in km)** | 1.0 | 1.0–1.0 | 0.041 | 0.9 | 0.6–1.4 | 0.641 |
| **Gender** | | | | | | |
| Male | 0.6 | 0.2–1.4 | 0.208 | . | . | . |
| **Age groups in the last snakebite (years)** | | | | | | |
| <18 | 1 | . | . | . | . | . |
| 18–45 | 5.9 | 1.2–27.9 | 0.026 | 1.1 | 0.1–17.0 | 0.957 |
| 46–60 | 8.6 | 1.5–51.2 | 0.018 | 1.7 | 0.1–52.3 | 0.762 |
| ≥60 | 5.5 | 0.8–36.2 | 0.076 | 17.4 | 0.2–1322.8 | 0.195 |
| **Education (years of study)** | | | | | | |
| Illiterate | 1 | . | . | . | . | . |
| ≤4 | 0.7 | 0.3–1.5 | 0.316 | 2.8 | 0.6–13.6 | 0.204 |
| >4 | 0.4 | 0.2–0.8 | 0.007 | 2.4 | 0.6–8.9 | 0.211 |
| **Occupation** | | | | | | |
| Agriculture | 1 | . | . | | | |
| Fishing | 0.7 | 0.3–1.6 | 0.444 | 0.9 | 0.1–5.7 | 0.883 |
| Retired | 5.7 | 1.6–21.1 | 0.008 | 1.0 | 0.1–16.2 | 0.998 |
| Others | 0.5 | 0.2–1.4 | 0.212 | 0.3 | 0.04–1.7 | 0.167 |
| **Marital status** | | | | | | |
| Married/stable relationship | 1 | | | | | |
| Unmarried | 0.7 | 0.3–1.3 | 0.243 | | | |
| Divorced | 1.0 | 0.1–16.6 | 0.991 | | | |
| Widow | 2.0 | 0.2–23.0 | 0.566 | | | |
| **Monthly income (minimum wages)** | | | | | | |
| <1 | 1 | | | | | |
| 1–3 | 2.3 | 0.4–12.9 | 0.347 | | | |
| ≥3 | 0.6 | 0.05–6.4 | 0.651 | | | |
| **Income source#** | | | | | | |
| Pension | 3.5 | 1.3–9.4 | 0.014 | 0.4 | 0.02–8.4 | 0.543 |
| *Bolsa Família* | 0.6 | 0.2–1.9 | 0.422 | | | |
| Subsistence income | 0.6 | 0.3–1.2 | 0.133 | 0.2 | 0.02–1.6 | 0.125 |
| **Housing characteristics** | | | | | | |
| Wooden house | 0.4 | 0.04–5.0 | 0.514 | | | |
| Houseboats | 0.3 | 0.02–4.9 | 0.398 | | | |
| **Number of snakebites during lifetime** | | | | | | |
| 1 | 1 | . | . | 1 | | |
| 2 | 1.9 | 0.8–4.1 | 0.127 | 2.2 | 0.2–25.4 | 0.526 |
| ≥3 | **5.2** | **1.8–15.1** | **0.002** | | | |
| **Date of the last snakebite#** | | | | | | |
| <3 months | 1 | . | . | 1 | . | . |
| 3–6 months | . | . | . | . | . | . |
| 6 months-1 year | 0.6 | 0.5–5.2 | 0.608 | 0.3 | 0.01–9.4 | 0.505 |
| 1–5 years | 1.4 | 0.2–7.9 | 0.719 | 3.5 | 0.2–60.7 | 0.393 |
| 6–10 years | 1.7 | 0.2–11.6 | 0.605 | 2.5 | 0.1–77.6 | 0.592 |

*(Continued)*

**Table 3.** (*Continued*)

| Variable | OR | 95%CI | *p* | aOR | 95%CI | *p* |
|---|---|---|---|---|---|---|
| ≥10 years | 3.7 | 0.7–20.2 | 0.128 | 4.2 | 0.2–72.8 | 0.319 |
| **Place where snakebite occurred** | | | | | | |
| Agricultural land | 1 | . | . | | | |
| Household area | 0.9 | 0.4–2.1 | 0.859 | | | |
| Trails to access workplaces | 1.4 | 0.4–4.5 | 0.568 | | | |
| River margins | 1.3 | 0.5–3.4 | 0.562 | | | |
| Rubber plantation | 1.9 | 0.6–6.9 | 0.290 | | | |
| Açaí plantation | 0.6 | 0.05–7.2 | 0.698 | | | |
| Game hunting | 1.2 | 0.07–20.8 | 0.887 | | | |
| **Use of individual protection** | | | | | | |
| Yes | 1.0 | 0.4–2.4 | 0.922 | | | |
| **Type of envenomation** | | | | | | |
| *Bothrops* | 1 | . | . | 1 | . | . |
| *Lachesis* | 2.5 | 1.2–5.1 | 0.015 | 2.1 | 0.6–7.8 | 0.247 |
| *Micrurus* | 2.8 | 0.3–32.0 | 0.401 | 20.1 | 0.6–700.3 | 0.098 |
| **Anatomical region of the bite** | | | | | | |
| Lower limbs | 1 | . | . | 1 | . | . |
| Upper limbs | 0.5 | 0.2–1.2 | 0.124 | 0.3 | 0.04–1.4 | 0.121 |
| **Use of traditional medicine** | | | | | | |
| Yes | 9.9 | 4.5–21.7 | <0.001 | **11.6** | **3.4–39.8** | **<0.001** |

OR, 95%CI: Odds Ratio, with its respective 95% Confidence Interval. All variables associated with the outcomes at a significance level of $p<0.20$ in the univariate analysis were included in the multivariable analysis. Statistical significance was considered if $p<0.05$ in the Hosmer-Lemeshow goodness-of-fit test. aOR: Adjusted Odds Ratio.

healthcare networks. A lesser availability of financial resources to cover transportation may partly explain this difference. From a geographical point of view, this population is outside the territorial healthcare network not just because of the great distance from urban centers, but also due to the difficulties in navigating of the rivers. Most communities that were studied are located on the banks of rivers that meander greatly, especially the Juruá River, which is considered the most meandering river in the basin. This river is extremely important for the region, and serves as a fluvial highway for several communities, since roads are non-existent in most of this territory. However, the fact that the river is very sinuous makes the traffic very slow along its entire length. In addition, in the low water period, medium and large vessels have to suspend operations, since they are unable to travel along this river due to the presence of sand banks. In this study, among participants who had access to healthcare, 34.5% sought it after more than 6 hours, and 12.1% of the participants took >72 hours to receive medical care. As a medical emergency, snakebite envenomations require antivenom treatment without delay, preferably within the first 6 hours after the bite [12]. Late treatment may result in greater severity and higher rates of case fatality [11]. In cases that involve vascular trauma, late medical assistance was also associated with case severity, longer hospitalization time, and greater probability of limb amputation [41]. Indeed, the distribution of emergency services in Brazil does not facilitate access by the population due to the geographical barriers associated with great distances, with the poorest access occurring in rural areas and the Amazon region [42].

Although attempts to improve access to health services through the use of fluvial medical units have been carried out in the Amazon, effectiveness was limited from the users' perspective, since the dispersed housing pattern in riverside communities requires the locomotion of

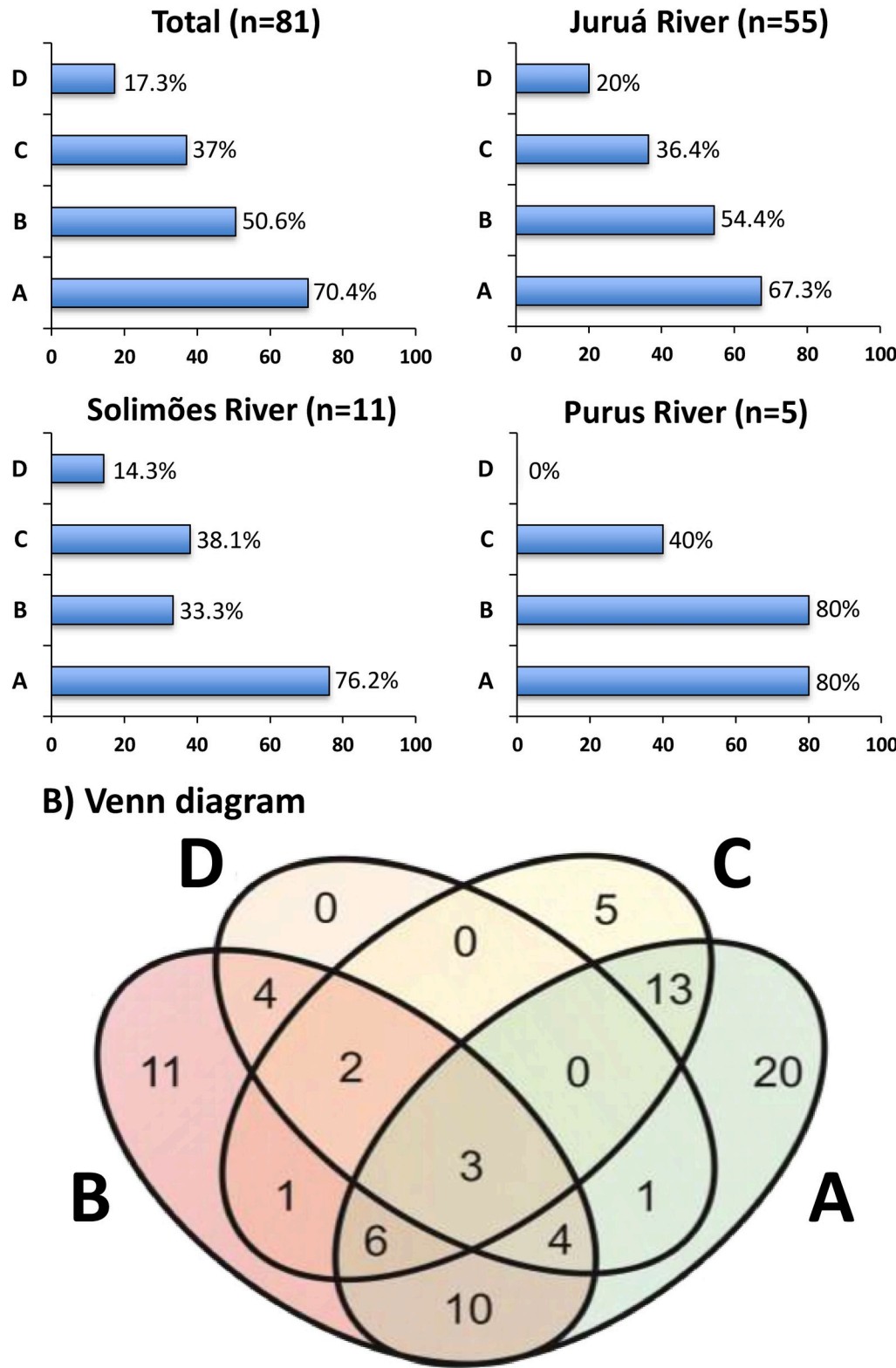

## A) Reasons for not accessing healthcare

**Total (n=81)**

- D 17.3%
- C 37%
- B 50.6%
- A 70.4%

**Juruá River (n=55)**

- D 20%
- C 36.4%
- B 54.4%
- A 67.3%

**Solimões River (n=11)**

- D 14.3%
- C 38.1%
- B 33.3%
- A 76.2%

**Purus River (n=5)**

- D 0%
- C 40%
- B 80%
- A 80%

## B) Venn diagram

**Fig 4. Reasons given by the participants for not to seek heathcare. A**) Reasons for not accessing healthcare and comparison of between study areas. **B**) Venn diagram showing the numbers of participants with their respective reason for not accessing healthcare and overlap of reasons given by the participants. A) Patient prioritized traditional treatments, relying on their effectiveness; B) Failure to recognize the situation as being potentially serious; C) Lack of financial resources and means of transport; D) Resistance to seek medical assistance, despite family pressure.

**Table 4. Characteristics of the four deaths from snakebites as described by family members.**

| Case | Locality | Description¶ |
|------|----------|--------------|
| 1 | Community of 3 Unidos, municipality of Eirunepé, Juruá River banks | An elderly man, 67 years old, farmer and fisherman, married, was bitten by a "jararaca" (possibly *Botrops atrox*), popularly known as 'jararaca', while planting manioc thirty minutes from his residence. When he saw the snake, he decided to kill it with a machete and, before it received the blow, the snake struck the patient's foot. In great pain, the man was carried into the house by companions. With little mony available to travel to the city, he chose to use only traditional medicines to 'treat the effects of the envenomation' (*Específico Pessoa* and the 'second step tea'). Family members reported that for four days the man had severe pain, persistent bleeding at the site of the bite, hematuria and extensive edema in the lower limb. The victim died ~7 days after the bite, without medical assistance. |
| 2 | Community of Taboca, municipality of Tapauá, Purus River banks | An 18-year-old man, farmer and fisherman, suffered a snakebite during his morning fishing activity. Soon after arriving at the river, he was bitten on the foot. The man reported that he had been bitten by the snake known as 'surucucurana' (possibly *Bothrops atrox*). Upon arriving back at his house, a health worker who was visiting the community was immediately called to take the victim to a hospital in the municipality of Tapauá, 7 hours away, using a motorboat provided by the municipality. During the journey, the patient had severe bleeding, pain and also passed out. Upon arriving at the referral unit, the victim was informed that no antivenom was available. The patient remained in the unit until he was informed of a transfer to the referral unit in Manaus, but before starting the journey he died. |
| 3 | Community of Novo Horizonte, municipality of Guajará, Juruá River banks | A three-year-old girl, resident of a community on the banks of the Juruá Riverwas, according to her father, playing in the backyard when a big snake identified by the family member as a 'surucucu pico-de-jaca' (the bushmaster *Lachesis muta*) bit the child's back. The father reported that it looked like the child had been pushed forward. After the bite, the child immediately reported a excruciating pain at the site of the bite and finally passed out. Family members reported that the child could not stand 30 minutes, and there was not enough time to seek healthcare. The child died at home and was buried in the backyard of the house, with no official record of death. |
| 4 | Community of 3 Unidos, municipality of Eirunepé, banks of the Juruá River | An 81-year-old man, farmer and fisherman, went fishing near his home. His family reported that he took a trail to get to the fishing site. Upon reaching the banks of this watercourse, he passed over a trunk and reported being bitten on the leg by a 'jararaca' (possibly *Botrops atrox*). Soon after the accident, he refused to seek healthcare because he 'had suffered another snakebite years ago and survived'. After ~5 days days, the condition worsened with the presence of myalgia, renal failure with dark-colored urine (the color of coca-cola), pain, and edema in the limb. The wife reported that the clinical situation worsened and he later died. |

¶ After indication by the informant, family members were invited to be interviewed for collection of information on deaths.

users to reach the mobile unit's services, as well as when seeking specialized care in the urban centers [39]. Generally, riverine population travel to health services by small canoes, which can navigate the long distances between remote areas to communities with more capacity for assistance. Services, such as transport of the injured patient, even when triggered by the health network, may not reach the riverside communities since the ambulance service motorboats do not travel long distances from their base, which leaves remote areas without coverage. In addition, it is important to point out that in the Amazon, a region with continental dimensions, patients are not limited to seeking health services within the geographical limits of the municipality. For the patients of the Juruá River, for example, it may be more feasible to seek for care at the municipality of Cruzeiro do Sul (state of Acre) in search of antivenoms as opposed to travelling to Manaus (state of Amazonas), a city with greater installed capacity and more technologically advanced in the health sector. Another example is the patients of the Solimões River who, also due to greater viability, choose to seek health care available in neighboring countries (Colombia or Peru), thus reducing the financial costs involved in transport and the time taken, which is a determinant variable for the good evolution of cases.

In Brazil, snakebites are compulsorily recorded by the Brazilian Communicable Diseases Surveillance System [*Sistema de Informação de Agravos de Notificação* (SINAN)] on data report forms used in the investigation and follow-up of cases in the hospitals [43]. Data collected from the hospitals are transferred electronically to reach the central level at the Ministry of Health and, in response, antivenom is sent from the central to the local level in an amount estimated by the number of reported cases [15]. Failure to report snakebites to the relevant authorities ends in a high rate of underreporting as observed in this study, and creates a bias when interpreting snakebite statistics, with negative consequences for the effectiveness of public policies, since decisions regarding antivenom distribution and the misallocation of other resources are based on official reports. The estimate of the magnitude of underreporting given by this study (almost 50%) should help health policy makers to design more appropriate snakebite management strategies and perform better cost-benefit analyses. Recently, some studies have proposed that an optimized decentralized antivenom delivery framework using primary care facilities in the Amazon region would shorten the time between diagnosis and treatment and, as a result, improve the prognosis of snakebites [16]. We expect that maximizing access to antivenom treatment in these areas is likely to result in a decrease in underreporting and an increase in demand for antivenom, and therefore this process must be coordinated between all levels of the supply chain and the producers.

The case-fatality rate from snakebites is very uneven across the territory of the Brazilian Amazon, with 0.4% in the Manaus region and 1.5% in regions inhabited by Amerindian populations [15]. In this study, we found a general case-fatality rate of 2.3%, reaching 3.7% in the Juruá River communities, which confirms the worst situation of healthcare in these locations. This fact suggests that deaths from snakebites are quite common in remote areas of the Brazilian Amazon region. Only one patient sought medical assistance with the help of a health worker who was visiting the community using a motorboat that belongs to the municipality, but the family reported that upon arriving at the referral unit, antivenom was not available. Underreporting of deaths to the official mortality system was estimated at ~30%, which represents a limitation for accurately evaluating public health and estimating the economic burden of the problem [11]. In the Amazonas state, the analysis of 127 deaths from snakebites has shown that 22% died without medical assistance, 46.5% did not receive antivenom and 63.3% received incomplete treatment [11]. Of the four fatal snakebites found in the present study, two of them occurred with elderly patients (over 60 years old), which was identified as a risk factor for complications and death in snakebites, and which is probably due to the presence of comorbidities in this age group [12,44]. These two patients did not seek the health service

despite worsening symptoms, and death occurred after 5 to 7 days. Three of the four reported deaths showed signs of systemic bleeding. In a previous study, systemic bleeding, circulatory shock, sepsis and acute respiratory failure were factors that were observed to be strongly associated to fatal outcomes in the Amazon [11].

## Self-care practices from an ethnopharmacological perspective

In this study, we observed a high frequency and a wealth of practices, mostly represented by plant-derived medicines, were used to treat snakebites by riverine populations. Interestingly, practices varied enormously between river basins, confirming that the traditional therapeutic arsenal has some geographic and cultural specificity. The participants cited the prioritization of traditional treatments as the most common reason for not accessing healthcare facilities. The belief in the effectiveness of traditional practices delays the decision to seek the health service and, as observed in this study, was a factor that was significantly associated to the lack of access to healthcare. The use of traditional self-care practices are often recorded around the world as the cause of late medical assistance and poor prognosis in snakebites [5,18,19].

Few ethnopharmacological studies have been carried out in Amerindian and riverine groups within the Brazilian Amazon. Lack of interdisciplinary training programs, including different fields such as botany, chemistry, pharmacology and anthropology, research funding, and defined methods have contributed to the marginal status afforded to ethnopharmacology by the scientific community [45]. In general, ethnopharmacological investigations in the Brazilian Amazon have focused mostly on the use of medicinal plants [46]. Some reports show that caboclos demonstrate extensive knowledge and utilization of medicinal plants and make greater use of secondary forest, non-native and cultivated garden species than Amerindian populations [45,46]. It was observed that preparations using leaves, roots of the açaí palm tree (*Euterpe precatoria*) were used in two river basins (Solimões and Juruá Rivers), but with a much greater frequency and variety of preparation methods in the Juruá River communities. The fruits of açaí palm tree are used for the preparation of a purple juice, for consumption as a drink, and its roots, red in color, are used as traditional folk medicine for the treatment of snakebite and malaria [47]. To the best of our knowledge, however, there is no evidence in the literature about the efficacy of this plant species in neutralizing snake toxins. Some studies show that mnemonic processes are used to identify medicinal plants based in the relation of a color and a health disorder. Red plant substances, for example, are thought to advance the healing of skin and hemostatic disorders [48]. The use of açaí preparations is not explained merely by the availability of this species, since it is very common throughout the whole Amazon region. Depending on the geographical location of the victim, the use of traditional medicine may be more widespread to the detriment of communities that are closer to the urban area, which have greater possibility of access to antivenom. In addition, sociocultural influences seem to interfere directly in decision-making for treatment with antivenom, but at the same time it draws attention that traditional medicine practices do not follow a homogeneous pattern of treatments among riverine people in the different river basins. This fact may be the result of different factors, such as, for example, local biodiversity, the predominant indigenous ethnic diversity in each basin, with its socio-cultural specificities and even the interaction with transboundary populations in some cases in the Brazilian Amazon. In the Amazon, regional variation in plant-based treatment practices has also been observed for malaria [49,50].

As observed in other regions of the Amazon, the *Específico Pessoa* preparation was commonly used by participants living in communities of the three rivers, on its own or in combination with a series of other preparations (5,13). Preliminary studies have not found any effectiveness of *Específico Pessoa* in neutralizing the main activities of the *Bothrops atrox*

venom [51]. This preparation is not marketed in pharmacies, as it does not have approval from the health regulatory authorities, but it can be purchased in stores that sell veterinary products and even in small food and houseware markets. Indeed, many kinds of pharmaceuticals are easily available in floating markets and even from 'traditional healers'. In this study, some participants reported the use of conventional medicines, especially intravenous benzyl-penicillin and painkillers, in combination with homemade preparations. Situations of pharmaceutical pluralism are often observed in developing countries, with similarities or contrasts between Western and traditional medicines. In general, conventional medicines are understood as being fast-working and potent, which are characteristics that make them suitable for the relief of acute disorders, such as pain from snakebites; otherwise, traditional medicines are perceived as slower and milder, better for chronic and recurrent conditions, and without the strong side effects that are associated with conventional medicines [48]. Until the 1990s, there were no restrictions on the purchase of antibiotics in Brazil, and it was common for families to have a small stock of this class of drugs in their homes, even in injectable forms, as benzylpeni-cillin. In most of the snakebites in the remote Amazon, self-medication is the most common way of using conventional medicines and these are commonly purchased from unauthorized drug vendors. On the other hand, this type of access to pharmaceuticals provides greater personal autonomy since it enables people to treat themselves individually within the context of exclusion from the health network [48].

Some foods, such as cow's milk and tortoise broth, were also used for therapeutic purposes. In this study, it was also observed that some preparations, normally classified in the food category, such as milk or game meat, were cited by the participants as being resources used in the treatment of snakebites. The distinction of these categories—medicines or foods—are in many cases clearly distinguished for the same preparation (although commonly interrelated through metaphors of nutrition and strengthening the body to fight the disease), so that medicines and foods have different meanings and are prepared, applied and consumed with different intended outcomes [48]. The use of black stones and other topical procedures was also common, and were generally interpreted as a possible way to extract the venom inoculated at the bite site [52]. However, the efficacy of some practices adopted by the study participants is problematic to understand within the biomedical structure of knowledge. For example, we cite the use of the 'second step tea', prepared from the material scraped from the surfaces of the second step of a wooden staircase, which generally gives access from the river to the community, or mosquito infusions, in the Juruá communities. The production of a beneficial effect on a snakebite, from the use of these practices, must be understood in a cultural context, which requires an in-depth examination of the processual nature of healing and proximate outcomes that people in these situations really expect, beyond their chemical properties. If the therapeutic reputation of these practices has only death without disabling sequelae as the final outcome expected by patients, and secondarily the mitigation of suffering during the healing process, it is possible that these therapeutic strategies are reputed to be sufficiently effective. It merits note that the failure of a traditional medicine to produce cure does not necessarily undermine one's trust in the folk health system, but that the medicine and the individual were not suited in that particular instance, and an alternative medicines or 'more faith' would be necessary to yield the desired results [53]. In other cases, which become complicated and are transported to hospitals, having negative outcomes with medical care, even if late, the responsibility for ineffective treatment may be imputed to the official health system.

## How to take better care of yourself in conditions of vulnerability?

Failure to recognize the situation as being potentially serious was reported by half of the participants who did not seek medical care. Previous experiences with mild envenomations, bites by

non-venomous snakes or dry bites (treated or not by traditional care) could generate a false perception of cure and, thus, in subsequent snakebites, blur the perception of emergency in face of a severe envenomation, leaving the victim comfortable about not seeking healthcare services. Actually, our group has previously shown that a number of patients sought help only after warning signs, such as the onset of unbearable pain, disfiguring edema, bleeding and decreasing functional mobility [5]. Many riverside residents need to use their own financial means to reach health services. However, in some cases, their income is not enough for the displacement. In the municipality of Manaus, patients also reported financial difficulties that stemmed from the need to acquire fuel for cars or boats (either their own or borrowed), and/ or expenses involving taxis, medicines and food during the journey [5]. Since victims of snakebites are generally economically vulnerable, reducing costs for the patients by cash rewards or refunds may be a viable measure. In addition to all these factors, there may also be a fear of leaving the family and their farms, as the cattle, pets and assets depend on their care, and, generally, they have no close friends or relatives who can do it in their absence. Another issue related to dependence on the environment refers to the rural worker, who has a subsistence income that comes directly from his daily labor in order to feed his family, whether through agriculture, forestry, hunting or fishing.

## Limitations

The collection of information in this study depended on the memory of the interviewees, who in many cases were already elderly or had suffered snakebites that had occurred more than 10 years ago. Leveling and sharpening biases are also possible, with memory distortions introduced by the loss of details in a recollection over time. These are often concurrent with a selective recollection of certain details that take on exaggerated significance in relation to the details or aspects of the experience that are lost through leveling. For example, information about the season in which the snakebite occurred would be very relevant to understanding differences in access to healthcare. Unfortunately, such detailed information was not added to the data collection instrument, since research participants would be unable to recall this information, as a large number of cases occurred many years ago. Another possible limitation of this study is regarding a sampling bias since the informants identified might not have knowledge about all the other cases of snakebites. This means a researcher might not be able to uncover all cases and deaths. Moreover, some communities were not approached due to difficulties in mooring the vessel on the riverbank, preventing the study from having a greater number of locations. In addition, unfortunately, the number of inhabitants per community was not obtianed, which did not allow the calculation of the prevalence of snakebites in this population.

## Concluding remarks

In the last decade, the Brazilian government has implemented some organizational arrangements to improve the health care of the riverine population, however, when it comes to the need for immediate attention, as in the case of snakebite, these policies still seem to be insufficient to avoid complications, sequelae, and deaths. There is a growing body of literature on the treatment practices of snakebite victims who have not consulted any healing specialist for their illness, and it appears as if most cases are initially treated at home using traditional medicines or pharmaceuticals and a great proportion of these cases never comes to the attention of a medical practitioner. As a result, the major finding of this study is the unexpectedly high underreporting of snakebite cases and associated deaths in communities of three river basins in the western Brazilian Amazon, which resulted from poor access to healthcare services. In this context, almost half of the participants reported not having sought a hospital, and those

who sought medical care did so with delay. Snakebite victims utilized three main different healing systems: 1) self-care using miscellaneous techniques; 2) official medical healthcare generally combined with traditional practices; and 3) self-care using traditional practices combined with conventional medicines. Geographical and income barriers end in inequalities in healthcare and, possibly, collaborate so that riverine residents prioritize traditional treatment, although the knowledge about the use of natural resources of the forest for healing and their customs also enjoy great importance in this sense. However, it is not possible to say that riverine residents would discard traditional treatments if they had ample access to healthcare services, or whether they would combine both treatments. To mitigate burden of snakebites in the Brazilian Amazon, decision-makers should focus on the implementation of an innovative intervention that optimize timely care delivery, and includes antivenom distribution among existing community healthcare centers.

## Supporting information

**S1 File. STROBE checklist.**
(DOC)

**S2 File. Communities and their geographic locations, with at least one participant who claimed to have been bitten by a snake.**
(XLSX)

**S3 File. Traditional medicines used by the study participants.**
(DOCX)

**S4 File. Study raw data.**
(XLSX)

## Acknowledgments

We would like to express our thanks to the 9th Command of Naval District (Brazilian Navy, Manaus), and to Mrs. Natalia Salinas, scientific research manager at the Institute of Weights and Measures (IPEM/INMETRO-AM, Manaus), for the valuable support on the research trips.

## Author Contributions

**Conceptualization:** Fernando Val, Marcus Lacerda, Fan Hui Wen, Wuelton Monteiro, Jacqueline Sachett.

**Data curation:** Vinícius Azevedo Machado, André Sachett, Jacqueline Sachett.

**Formal analysis:** Alexandre Vilhena Silva-Neto, Vinícius Azevedo Machado, Vanderson Souza Sampaio, André Sachett, Wuelton Monteiro.

**Funding acquisition:** Marcus Lacerda, Wuelton Monteiro, Jacqueline Sachett.

**Investigation:** Guilherme Kemeron Maciel Salazar, Joseir Saturnino Cristino, João Arthur Alcântara, Wuelton Monteiro, Jacqueline Sachett.

**Methodology:** Fan Hui Wen, Wuelton Monteiro, Jacqueline Sachett.

**Project administration:** Altair Seabra Farias, Wuelton Monteiro, Jacqueline Sachett.

**Resources:** Wuelton Monteiro, Jacqueline Sachett.

**Software:** Alexandre Vilhena Silva-Neto, André Sachett.

**Supervision:** Altair Seabra Farias, Vinícius Azevedo Machado, Wuelton Monteiro, Jacqueline Sachett.

**Validation:** Vinícius Azevedo Machado, Felipe Murta.

**Visualization:** Guilherme Kemeron Maciel Salazar, Joseir Saturnino Cristino, Alexandre Vilhena Silva-Neto, João Arthur Alcântara, Paulo Sérgio Bernarde, Wuelton Monteiro.

**Writing – original draft:** Altair Seabra Farias, Wuelton Monteiro.

**Writing – review & editing:** Felipe Murta, Vanderson Souza Sampaio, Fernando Val, Paulo Sérgio Bernarde, Marcus Lacerda, Fan Hui Wen, Jacqueline Sachett.

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
