## [Decision Letter · Decision Letter 0]

15 Jul 2021

Dear Dr. Monteiro,

Thank you very much for submitting your manuscript "Snakebites in “Invisible Populations”: A Cross-Sectional Survey in Riverine Populations in the Remote Western Brazilian Amazon" for consideration at PLOS Neglected Tropical Diseases. As with all papers reviewed by the journal, your manuscript was reviewed by members of the editorial board and by several independent reviewers. The reviewers appreciated the attention to an important topic. Based on the reviews, we are likely to accept this manuscript for publication, providing that you modify the manuscript according to the review recommendations. 

Sincerely,

Indika Gawarammana

Deputy Editor

Indika Gawarammana

Deputy Editor

Reviewer's Responses to Questions

**Key Review Criteria Required for Acceptance?**

**Methods**

-Are the objectives of the study clearly articulated with a clear testable hypothesis stated?

-Is the study design appropriate to address the stated objectives?

-Is the population clearly described and appropriate for the hypothesis being tested?

-Is the sample size sufficient to ensure adequate power to address the hypothesis being tested?

-Were correct statistical analysis used to support conclusions?

-Are there concerns about ethical or regulatory requirements being met?

Reviewer #1: The manuscript is well written and clearly shows the importance to know the magnitude of snakebite underreporting has to the healthy system constituting a serious problem and an aspect not addressed in a detailed form in the literature like this manuscript does it. 

The objectives are clear and the design appropriate, but I would like to know if the authors have the answers about the quantity of envenoming that occurs in the distinct seasons and if the floods river have some importance to this quantity. I think that this aspect was not mentioned and has some implications for the results obtained. In the discussion section, the authors mentioned something in an "en passant" manner. The methods are clear and appropriate for the study proposed.

Reviewer #2: (No Response)

**Results**

-Does the analysis presented match the analysis plan?

-Are the results clearly and completely presented?

-Are the figures (Tables, Images) of sufficient quality for clarity?

Reviewer #1: Results obtained are new and with an importance to the field. Snakebite underreporting is a real problem not only in Brazil but in other countries too. This aspect has many consequences to the health system that must be raised and debated for improvement. 

It is important to mention or comment about the season of the year where the authors found the highest accidents, if it occurs in a specific season or not. Another aspect that can be addressed is the floods river and if this aspect impacts the numbers of accidents or not in these specific studied areas. 

The authors conducted some health education during this trip with this invisible population?

Reviewer #2: (No Response)

**Conclusions**

-Are the conclusions supported by the data presented?

-Are the limitations of analysis clearly described?

-Do the authors discuss how these data can be helpful to advance our understanding of the topic under study?

-Is public health relevance addressed?

Reviewer #1: The conclusions is supported by the results obtained with an importance to the field. Snakebite underreporting is a real problem not only in Brazil but in other countries too. This aspect has many consequences to the health system that must be raised and debated for improvement.

Reviewer #2: (No Response)

**Editorial and Data Presentation Modifications?**

Reviewer #1: No. Accept.

Reviewer #2: (No Response)

**Summary and General Comments**

Reviewer #1: The manuscript must have to have page numbers as well as line numbers to the reviewers' scores are properly corrected. 

It is important to mention or comment about the season of the year where the authors found the highest accidents, if it occurs in a specific season or not. Another aspect that can be addressed is the floods river and if this aspect impacts the numbers of accidents or not in these specific studied areas. 

The authors conducted some health education during this trip with this invisible population?

Reviewer #2: Thank you for the opportunity to review this important cross-sectional study of snakebite in the Riverine populations of the Amazon. This is important research that gives us important information regarding under reporting of snakebite in this population. please consider the suggestions below:

1. The manuscript is well written and the language flows well, but I do have a critique regarding its length. There are many areas of the manuscript that are not essential to the understanding of this specific study. I feel that some of the topics would be better served as publication a review article as opposed the this specific study. Examples include:

Methods Study sites. The first paragraph is certainly informative, but goes into a lot of detail that is important to snakebite in the Amazon generally, but not important to this specific study. Especially the part about the flooding, sedimentation, suspended solids, etc. Also the paragraph b beginning with "These communities present a model of land occupation..." Once again, this is important information but much to detailed to directly inform this study. I strongly suggest a separate review manuscript for this discussion. The length of the discussion is also too long for the reader to maintain focus on the primary study question. I would suggest removing almost all of the infectious disease part unless it informs snakebite. Only the parts to pertain directly the the under-reporting and the lack of access to healthcare should be retained for this manuscript. The conclusion can likewise be shortened. 

2. Study design. This study is described as a quantitative study, yet there seems to be elements of a qualitative study as well. I cannot be certain from the methods described, but I would request clarity to determine if this is mixed methods or solely quantitative. An example of my uncertainty is the sentence "The inclusion of new participants finished once the saturation point had been reached." please describe "saturation". do you mean thematic saturation as is accepted in qualitative methods. If not is there another way the sample size was determined. Additionally the questionnaire had closed and open ended questions. Were the open ended questions used to determine thematic saturation. This can likely be easily addressed by giving a little more detail.

3. As this article will be read by an international audience, I would like to point out some language that will be culturally troublesome for at least some the your readers. I strongly recommend that the word "Negro" be removed. This term is very offensive in the US. I also suggest removing the majority of the second paragraph that discusses genetics as it isn't vital to introduction or the study. Simply describe the population as admixed and remove the "genetic" part. To my knowledge we have no evidence that there is any important genetic difference between these various populations that impact snakebite outcomes. The current language unintentionally contributes to the paradigm of seeing these people as "other". Since this isn't a genetic study, I strongly suggest just removing.

PLOS authors have the option to publish the peer review history of their article (what does this mean?). If published, this will include your full peer review and any attached files.

Reviewer #1: No

Reviewer #2: No

Figure Files:

Data Requirements:

Reproducibility:

References

---

## [Editor Report · Decision Letter 1]

26 Jul 2021

Dear Dr. Monteiro,

Thank you very much for submitting your manuscript "Snakebites in “Invisible Populations”: A Cross-Sectional Survey in Riverine Populations in the Remote Western Brazilian Amazon" for consideration at PLOS Neglected Tropical Diseases. As with all papers reviewed by the journal, your manuscript was reviewed by members of the editorial board and by several independent reviewers. The reviewers appreciated the attention to an important topic. Based on the reviews, we are likely to accept this manuscript for publication, providing that you modify the manuscript according to the review recommendations. 

Sincerely,

Indika Gawarammana

Deputy Editor

Indika Gawarammana

Deputy Editor

Figure Files:

Data Requirements:

Reproducibility:

References

---

## [Editor Report · Decision Letter 2]

24 Aug 2021

Dear Dr. Monteiro,

We are pleased to inform you that your manuscript 'Snakebites in “Invisible Populations”: A Cross-Sectional Survey in Riverine Populations in the Remote Western Brazilian Amazon' has been provisionally accepted for publication in PLOS Neglected Tropical Diseases.

Best regards,

Indika Gawarammana

Deputy Editor

Indika Gawarammana

Deputy Editor

---

## [Editor Report · Acceptance letter]

2 Sep 2021

Dear Dr. Monteiro,

We are delighted to inform you that your manuscript, "Snakebites in “Invisible Populations”: A Cross-Sectional Survey in Riverine Populations in the Remote Western Brazilian Amazon," has been formally accepted for publication in PLOS Neglected Tropical Diseases.

Best regards,

Shaden Kamhawi

co-Editor-in-Chief

Paul Brindley

co-Editor-in-Chief
